# Detection of Primary DNA Lesions by Transient Changes in Mating Behavior in Yeast *Saccharomyces cerevisiae* Using the Alpha-Test

**DOI:** 10.3390/ijms241512163

**Published:** 2023-07-29

**Authors:** Anna S. Zhuk, Anna A. Shiriaeva, Yulia V. Andreychuk, Olga V. Kochenova, Elena R. Tarakhovskaya, Vladimir M. Bure, Youri I. Pavlov, Sergey G. Inge-Vechtomov, Elena I. Stepchenkova

**Affiliations:** 1Institute of Applied Computer Science, ITMO University, 191002 St. Petersburg, Russia; 2Vavilov Institute of General Genetics, St. Petersburg Branch, Russian Academy of Sciences, 199034 St. Petersburg, Russia; elena.tarakhovskaya@gmail.com (E.R.T.); ingevechtomov@gmail.com (S.G.I.-V.); 3Laboratory of Amyloid Biology, St. Petersburg State University, 199034 St. Petersburg, Russia; yullinnabk@yandex.ru; 4Department of Genetics and Biotechnology, St. Petersburg State University, 199034 St. Petersburg, Russia; annabiologic@gmail.com (A.A.S.); olga_kochenova@hms.harvard.edu (O.V.K.); 5Howard Hughes Medical Institute, Department of Biological Chemistry and Molecular Pharmacology, Harvard Medical School, Blavatnik Institute, Boston, MA 02115, USA; 6Department of Plant Physiology and Biochemistry, St. Petersburg State University, 199034 St. Petersburg, Russia; 7Faculty of Applied Mathematics and Control Processes, St. Petersburg State University, 199034 St. Petersburg, Russia; vlb310154@gmail.com; 8Eppley Institute for Research in Cancer, Fred and Pamela Buffett Cancer Center, the University of Nebraska Medical Center, Omaha, NE 68198, USA; ypavlov@unmc.edu; 9Departments of Biochemistry and Molecular Biology, Microbiology and Pathology, Genetics Cell Biology and Anatomy, the University of Nebraska Medical Center, Omaha, NE 68198, USA

**Keywords:** DNA lesions, temporary phenotype modifications, mutations, DNA repair, alpha-test

## Abstract

Spontaneous or induced DNA lesions can result in stable gene mutations and chromosomal aberrations due to their inaccurate repair, ultimately resulting in phenotype changes. Some DNA lesions per se may interfere with transcription, leading to temporary phenocopies of mutations. The direct impact of primary DNA lesions on phenotype before their removal by repair is not well understood. To address this question, we used the alpha-test, which allows for detecting various genetic events leading to temporary or hereditary changes in mating type α→a in heterothallic strains of yeast *Saccharomyces cerevisiae*. Here, we compared yeast strains carrying mutations in DNA repair genes, mismatch repair (*pms1*), base excision repair (*ogg1*), and homologous recombination repair (*rad52*), as well as mutagens causing specific DNA lesions (UV light and camptothecin). We found that double-strand breaks and UV-induced lesions have a stronger effect on the phenotype than mismatches and 8-oxoguanine. Moreover, the loss of the entire chromosome III leads to an immediate mating type switch α→a and does not prevent hybridization. We also evaluated the ability of primary DNA lesions to persist through the cell cycle by assessing the frequency of UV-induced inherited and non-inherited genetic changes in asynchronous cultures of a wild-type (*wt*) strain and in a *cdc28-4* mutant arrested in the G1 phase. Our findings suggest that the phenotypic manifestation of primary DNA lesions depends on their type and the stage of the cell cycle in which it occurred.

## 1. Introduction

The genetic material in living cells changes because of replication errors and constant exposure to damaging agents. Free radicals, endogenous metabolites, and ionizing or UV radiation may target any atom or chemical bond in a DNA molecule [1,2]. Some common types of deviation from canonical DNA structure include mismatches, oxidized or methylated bases, bulky adducts, abasic sites, 6–4 photoproducts, pyrimidine dimers, cross-links, as well as single- and double-strand breaks (Figure 1) [3,4]. During repair, some types of DNA lesions may be converted into more severe lesions, for example, alkylated bases may be converted into double-strand breaks [5]. DNA lesions can block replication and gene expression, thus decreasing the physiological performance of the cell or even inducing cell death [6]. To cope with the consequences of DNA damage, cells use various DNA repair and temporary damage tolerance mechanisms [1,2,7,8]. The joint activity of several repair and replication enzymes restores DNA’s chemical structure, thus removing temporary barriers for replication and transcription machinery. Generally, the repair is very accurate, so in most cases, the initial DNA sequence is restored. However, in rare cases, DNA repair leads to mutations—hereditary changes in the DNA sequence. The removal of primary DNA lesions by recombination can also result in changes in genetic material, such as point mutations, chromosome rearrangements, and aneuploidy [9,10]. When repair systems are overwhelmed, mechanisms of damage tolerance are activated, one of which is error-prone TransLesion DNA Synthesis (TLS) [11]. During this process, specialized DNA polymerases bypass the lesions, creating a new nucleotide sequence that differs from the initial one. This process is highly mutagenic. Thus, inaccurate repair, recombination, or TLS on damaged DNA leads to the accumulation of point mutations and chromosome rearrangements, causing phenotype changes (Figure 1). Usually, we can detect the changes on the phenotype level only when the mutation is already established [12,13], though a cell can live a considerable time with damaged genetic material, as the repair process is not immediate. For example, the repair of double-strand breaks in yeast *Saccharomyces cerevisiae* takes up to 8 h, which is quite long considering that the yeast cell cycle lasts only two hours under optimal conditions [5]. So, it is plausible that primary lesions may affect cell phenotype before they become mutations (Figure 1).

There are only a few examples where the phenotypic expression of primary lesions was shown experimentally. Resnick and Holliday (1971) showed the temporary inactivation of nitrate reductase in *Ustilago maydis* after the UV-induced formation of cyclobutane dimers in the structural gene of the enzyme; subsequently, the enzyme activity was partially restored by photoreactivation [14]. A few studies showed that erroneous transcription over primary DNA lesions induced phenotypic changes in non-dividing cells. Thus, in *Escherichia coli* and mammalian cells, 8-oxoguanine (8-oxoG) and uracil incorporation into a transcribed strand of the luciferase reporter gene resulted in the production of aberrant transcripts, which could then be repeatedly translated to generate large numbers of proteins variants, inducing changes in luciferase activity—phenotype changes in transfected human cell lines or in transformed *E. coli* cells [15,16,17,18].

The first steps of the conversion of DNA lesions into mutations, including lesion dynamics before and during repair, and their impact on gene expression are not well known. The temporary nature of primary lesions and difficulties in discriminating lesions and mutations on the phenotype level hamper such studies. Most test systems consider the repair endpoints only—gene mutations and recombination and chromosome aberrations—while a large amount of the primary lesions are excluded from the analysis. 

To the best of our knowledge, the alpha-test is the only system that identifies different types of changes in DNA: point mutations, recombination, whole chromosome or chromosome arm loss, and primary lesions. This test measures the frequency of the mating-type switching from “α” to “a” in heterothallic *S. cerevisiae*, which is manifested by the formation of the illegitimate hybrids of two strains initially having the same mating type “α”. Under normal conditions, spontaneous illegitimate hybridization occurs at a frequency of 10^−5^–10^−6^ [19,20,21]. When cells are exposed to genotoxic agents, the frequency of mating-type switching increases, allowing for the measurement of the genetic activity of the tested factor. In the literature, various genotoxicity tests, based on the evaluation of the frequency of yeast mating-type switching have been described, such as the alpha-test, A-like faker assay, the test for illegitimate hybridization or cytoduction, and the nm-test [19,22,23,24]. They are all based on the same biological principles and represent variants of the same test system, with differences only in which genetic events affecting the mating type locus can be detected by a particular test system. The spectrum of detectable genetic events depends on the genetic markers carried by the tester strains. Depending on the research aims, it may be sufficient to evaluate the overall frequency of illegitimate hybridization using complementary markers allowing for the selection of illegitimate hybrids. When assessing the frequency of chromosomal abnormalities, it is necessary to use strains with genetic markers in both arms of chromosome III. Similarly, to distinguish between gene mutations and primary lesions at the *MAT* locus, it is important to use approaches and corresponding genetic markers that allow for the analysis of the haploid nucleus of the tester strain. 

The unique feature of the alpha-test is its ability to distinguish between inherited mutations and temporary DNA lesions. Even a transient change in mating type caused by a primary lesion may lead to the mating type switch followed by illegitimate hybrid formation. In addition to detecting temporary DNA lesions, the alpha-test also allows for studying the dynamics of primary lesion processing [25]. Two yeast cells can mate only when they are synchronized at the G1 phase of the cell cycle [26]. An illegitimate hybrid appearing on selective media indicates that the primary lesion manifests at the G1 phase, although the lesion could occur at any other stage. Therefore, it is possible to assess whether the lesion persists until the G1 stage using mutagens that act at certain stages of the cell cycle. Different variants of the alpha-test have been used previously to study the effects of DNA polymerases and chemical mutagens on genome stability [19,20,21,22,23,24,27,28,29,30,31]. Using the latest complete version of this test system, here we analyzed how primary DNA lesions affect cellular phenotype. Additionally, we assessed the ability of various primary lesions to pass through the cell cycle stages. We have obtained data confirming that the ability of primary lesions to influence the cellular phenotype largely depends on the chemical nature of the lesion.

## 2. Results and Discussion

### 2.1. The Strategy and Validation of the Test System

The alpha-test is the foremost approach used in this study to determine the types of DNA lesions that may affect the phenotype. The alpha-test is established on the principles of the mating type control in yeast *S. cerevisiae*. The mating type of *S. cerevisiae* cells is controlled by the *MAT* locus at the right arm of chromosome III [32,33]. Two cassettes (*HMRa* and *HMLα*) containing silent genetic information for the “a” and “α” types are located in the right and left arms of the same chromosome (Figure 2A) [34,35,36,37]. The *MAT* locus contains one of two alternative sequences—*MATa* or *MATα* (Figure 2A). The *MATα* locus has two open reading frames, *MATα1* and *MATα2*, which control the expression of a-specific (*asg*) and α-specific (*αsg*) genes: Matα1 is a positive regulator of *αsg* (not expressed without activation), while Matα2 is a negative regulator of *asg* (expressed constitutively in the absence of a repressor) [34,38,39,40]. Normally, mating-type switching happens via the “cassette mechanism”: the information from *HMRa* or *HMLα* replaces the content of the *MAT* locus by unidirectional conversion [41,42,43]. This kind of mating-type switching occurs in each budding cell of homothallic yeast strains, though it is a very rare event for heterothallic strains lacking HO endonuclease, which is necessary for the initiation of the conversion process [33,44,45,46]. The other mechanism of phenotypic mating type switch α→a is the inactivation of the *MATα* locus expression, where the absence of both *MATα* products leads to constitutive *asg* expression [25,29,47,48]. Thus, point mutations in the *MATα* locus, recombination or conversion between the *MATα* and the *HMRa* cassette, and the loss of chromosome III or the chromosome III right arm may lead to the heritable mating type switch. All these events can be registered in the alpha-test when two strains of the same mating type α are mixed under conditions selective for illegitimate hybrid growth. One of the two strains (the tester strain) is treated with genotoxic agents, inducing the mating type switch α→a. The second strain (the mating partner) bears recessive markers in both arms of chromosome III (*his4* and *thr4*). The use of such strains helps to determine whether the mating type switch was caused by the loss of chromosome III, loss of its right arm, either recombination or conversion between *MAT* and *HMRa* (Table 1; Figure 2B). Cells of the tester strain lacking a large fragment of chromosome III with the *MATα* locus may avoid death only by the hybridization with the mating partner because the lost genetic material in the hybrids is compensated by the copy from the second parent carrying recessive markers in chromosome III. The transient nature of primary lesions does not prevent their detection by the alpha-test but requires some modifications to the procedure (Table 1; Figure 2C). If the primary lesion is removed by repair after hybridization, the genetic material of the hybrid is still available for analysis, but it is a challenging task because hybrids contain the genetic material of both parental strains. To improve the testing capacity of the alpha-test, illegitimate cytoduction is used. The illegitimate cytoduction assay (a modified version of the alpha-test) allows for the isolation of the genetic material of the tester strain (Figure 2C). Cytoduction is an incomplete hybridization, where the cytogamy is not followed by nuclei fusion and diploid cell formation [22,49]. Cytoduction is highly stimulated by the *kar1*-*1* mutation [50,51,52]. The *kar1-1* mutant unable to fuse nuclei after mating (karyogamy-defective) was described in 1976 by James Conde and Gerald Fink. The authors showed that in the *kar1-1* mutant, the process of hybridization proceeds normally only until the moment of nuclear fusion, after which a multinuclear zygote (heterokaryon) is formed, which can produce cytoductants in subsequent divisions [51]. The *kar1-1* mutation (C448T or Pro150Ser) [52] is recessive and does not cause any obvious disruptions in the cell cycle, except for a block in karyogamy due to the disruption of the duplication of polar spindle bodies and defects in nuclear and cytoplasmic microtubules [52,53,54]. Presumably, the *kar1-1* mutation disrupts only one function of the protein—karyogamy. 

The resultant cytoductants are haploid cells with a mixed cytoplasm and a nucleus inherited from one of two parent cells (the recipient of the cytoplasm) (Figure 2C) [51]. To select cytoductants, it is necessary to use genetic markers with nuclear and cytoplasmic localization: the tester strain (recipient of the cytoplasm) should carry a recessive selective marker in the nucleus (e.g., cycloheximide resistance mutation *cyh^r^*) and should be devoid of mitochondria [*rho^0^*], while the mating partner strain (donor of the cytoplasm) should be respiration competent and sensitive to cycloheximide (*CYH^S^*). In this case, it becomes possible to select cells carrying a haploid nucleus from one strain and mixed cytoplasm from both parents on media containing cycloheximide and ethanol as the sole source of carbon. Analysis of the illegitimate cytoductants’ phenotype shows if repair was completed in an error-prone or error-free way. If DNA with a primary lesion is correctly repaired, cytoductants have the “α” mating type, and in a case of inaccurate repair, the illegitimate cytoductant changes its mating type. If the lesion is fixed as a mutation preventing the expression of both *MATα1* and *MATα2*, the mating type of the resulting cytoductant is recessive “a” (Alf phenotype). Sterile yeast cells, which cannot mate with either “a” or “α” cells (n/m phenotype), occasionally appear among illegitimate cytoductants. These clones bear a mutation in either *MATα1* or *MATα2* [24,34]. If *MATα2* is active and *MATα1* is not, *asg* are repressed and there is no *αsg* expression in the cell. In the case of *MATα2* inactivation, both *asg* and *αsg* are expressed simultaneously, and such cells may be regarded as analogs of heterozygous diploids *MATa*/*MATα*, which also cannot mate. Generally, the formation of these n/m cytoductants looks counterintuitive because a mutation in any *MATα* genes prevents hybridization. Apparently, it is possible only when primary lesions take place in both genes (Alf mating type), and after copulation, one lesion is repaired while the other is transformed into mutation [19,20,21].

Thus, illegitimate hybridization and cytoduction assays detect different types of genetic events (Table 1). Most of the genetic events scored in the alpha test affect the *MATα* locus, as strains carrying an extra copy of *MATα* on a plasmid have a more than 150-fold decrease in the frequency of illegitimate hybridization. Moreover, in sterile *matα2* mutants, the frequency of illegitimate hybridization with the α mating type strain is 10 times higher than in crosses between two strains of the α mating type [24]. Chromosome rearrangements and chromosome III loss are usually lethal for haploids and thus can be revealed only in illegitimate hybrids but not in cytoductants. At the same time, only the cytoduction test allows for distinguishing between mutations in the *MAT* locus and primary lesions (Table 1). Therefore, both variants of the alpha-test should be carried out simultaneously to obtain comprehensive information.

Since the frequencies of spontaneous illegitimate hybridization and cytoduction are known to be strain-specific [19,20], we determined these parameters in the yeast strains used in the study. The frequency of illegitimate hybridization was 2 × 10^−5^ and the frequency of illegitimate cytoduction was 10^−7^ (Table 2 and Table 3). These values correspond to the published data (10^−5^–10^−6^ for hybridization and 10^−7^–10^−8^ for cytoduction) [19,20]. The most frequent genetic changes scored in the illegitimate hybridization test are the loss of chromosome III (47%) or its right arm (38%), while reciprocal recombination between *MATα* and *HMRa* has the lowest frequency (0.5%) (Table 2).

Most of the illegitimate cytoductants (81%) have the “α” mating type (Table 3). Cytoductants of the “α” mating type appeared because of the temporary mating-type switching due to the inactivation of the *MATα* locus by a lesion followed by error-free repair after copulation. About 18% of cytoductants appeared due to the *HMRa* → *MATα* conversion, resulting in the “a” mating type (Table 3). Approximately 0.5% of cytoductants, having the recessive “a” (Alf) phenotype, emerged after the simultaneous mutations in the *MATα1* and *MATα2* genes or a mutation in the bidirectional promoter (Table 3). The rarest class of cytoductants (0.1%) are sterile due to mutation in *matα1* or *matα2* (Table 3).

As chromosome loss is lethal in haploids, we carried out additional experiments to test whether the loss of chromosome III would lead to immediate cell death and prevent illegitimate hybridization or if a yeast cell without chromosome III is able to mate. To distinguish two possibilities, we constructed a yeast strain in which the loss of chromosome III can be triggered by inducing transcription in the region of the chromosome centromere, as described before [55]. We inserted an inducible *pGAL1* promoter adjacent to the centromere DNA and transferred the cells onto a galactose media. Our results show that galactose-induced transcription of the centromere DNA led to a 5000-fold increase in the frequency of illegitimate hybridization (Figure 3A) in comparison to controls (the *wt* strain on glucose or galactose media and the *CEN3::pGal1* strain on glucose media). In the case of the *CEN3::pGal1* strain among several thousands of illegitimate hybrids selected on galactose media, only the class “chromosome loss” was detected. On the glucose media, the frequency of illegitimate hybridization and the proportions of different classes of genetic events in the strain-bearing *pGAL1* promoter did not differ from those in the wild-type strain (Figure 3B). Thus, our results clearly demonstrate that the loss of chromosome III does not lead to immediate cell death, and cells without chromosome III are able to mate.

### 2.2. Revealing the Phenotypic Expression of Different Primary DNA Lesions

It is known that different mutagens induce specific DNA lesions. We suggested that genotoxic factors causing such DNA lesions as double- and single-strand breaks, adducts, and abasic sites may be more effective inducers of illegitimate hybridization and cytoduction than mutagens that cause lesions with fainter disrupt DNA structure and weaker impede transcription and replication in the *MATα* locus. To check this suggestion, we used reference mutagens with known mechanisms of action (camptothecin and UV) and mutations causing defects in repair systems (*pms1* and *ogg1*). It is important to note that we used doses of mutagens and repair defects that are not lethal but still mutagenic for the used strains. In our experiments, the survival of *ogg1* and *pms1* strains as well as CPT-treated *wt* cells was close to 100%. In the case of UV, the survival of treated cells varied from 40 to 80%.

We used camptothecin (CPT) to study the phenotypic expression of double-strand DNA breaks. CPT is toxic to dividing cells via the inhibition of topoisomerase I, thus preventing DNA ligation and inducing the accumulation of single- and double-strand DNA breaks [56]. We have shown that CPT treatment significantly increases the frequency of illegitimate hybridization and cytoduction (~38-fold difference for hybridization and 6-fold for cytoduction) (Figure 4 and Figure 5). Phenotypic analysis of the illegitimate hybrids and cytoductants indicated that CPT is an effective inducer of all genetic changes scored in the alpha-test. In the illegitimate hybridization test, the frequency of CPT-induced chromosome III loss, chromosome III right arm loss, mutations and temporary lesions in *MATα*, and recombination and conversion between *MATα* and *HMRa* was 25–50 times higher than in the untreated strain (Figure 4). We concluded that a significant part of double-strand DNA breaks induced by CPT is fixed as heritable changes in genetic material. At the same time, CPT-induced lesions are able to provoke the nonheritable mating type switch α → a, since the frequency of illegitimate cytoductants of the α mating type is six times higher in the CPT-treated strain than in the untreated control (Figure 5). Thus, double-strand DNA breaks may have their own phenotypic expression; in particular, they may cause a transient mating type α → a switch in yeast.

To assess the phenotypic expression of mismatches, we inactivated the mismatch repair (MMR) system in yeast cells by introducing the *pms1* mutation in strains used in the alpha-test. The Pms1 protein is the principal regulator of MMR, and its inactivation leads to the accumulation of mismatched bases in the genomic DNA [57,58]. Here we have shown that disruption of the *PMS1* gene increased the frequency of illegitimate hybridization by 4-fold and induced chromosome III loss, the loss of chromosome III right arm, jointly counted mutations and temporary lesions in the *MATα,* and recombination between *MATα* and *HMRa* (Figure 4). At the same time, the inactivation of mismatch repair did not affect the frequency of illegitimate cytoduction (Figure 5A). The *pms1* mutants had an increased frequency of mutations in the *MATα* locus (in both or any of *MATα1* and *MATα2*) (Figure 5D–E) but a slightly decreased frequency of temporary lesions (Figure 5B). The absence of temporary lesion induction in the *pms1* mutants indicates that mismatches do not have the phenotypic expression in the alpha-test and the increased frequency of illegitimate mating in the *pms1* mutant is caused only by heritable genetic changes.

The *ogg1* mutants, accumulating endogenous 8-oxoG, were used to study the phenotypic expression of base modifications. Capable of canonical pairing with “C” and non-canonical pairing with “A” 8-oxoG induces transversion mutations [59,60]. In our study, the *ogg1* mutation increased the frequency of illegitimate cytoduction but not illegitimate hybridization (Figure 4A and Figure 5A). The disruption of base excision repair resulted in a 3-fold increase in temporary lesion frequency (Figure 5), a 3.7-fold increase in mutations in *MATα1* and *MATα2* (Figure 5D), and a 62-fold increase in mutations in any of *MATα1* or *MATα2* genes (Figure 5E). Thus, the alpha-test revealed the phenotypic expression of DNA templates containing 8-oxoG. Primary DNA lesions induced by UV light (cyclobutane dimers and 6–4 photoproducts) also showed phenotypic expression in the alpha-test. UV irradiation led to a considerable increase in illegitimate hybridization and cytoduction frequency in yeast cells and promoted all types of genetic events detected in the alpha-test (Figure 4 and Figure 5).

Our results indicate that primary lesions of different types affect the phenotype to varying degrees, which is likely related to the varying abilities of primary lesions to block transcription and replication. The primary lesions, such as mismatched bases, represent an incorrect pair of canonical nucleotides. However, the chemical structure of each DNA strand does not cause any blocking of transcription or replication. As a result, mismatches are expressed phenotypically in the alpha-test only if present in the transcribed strand and encode for an inactive mutated protein. A different situation is observed in the case of base modifications, double-strand breaks, or cyclobutane pyrimidine dimers and 6–4 photoproducts, which change the chemical structure of DNA. Primary lesions that significantly damage DNA structure naturally block replication and transcription, which disrupt the expression and transmission of genetic information through generations. We can conclude that the severity of DNA damage is one of the significant factors determining the ability of primary lesions to be expressed due to their ability to block template processes—transcription and replication.

### 2.3. Processing of the Primary DNA Lesions through the Cell Cycle

The alpha-test is unique in its ability to detect transient DNA lesions and has the potential to study the timing of lesion persistence and repair throughout the cell cycle. This is possible because the regulatory features of the mating type in *S. cerevisiae*, as well as the activity of different mutagens and repair systems, are timed to specific cell cycle phases [39,61]. In this study, we attempted to use the alpha-test to evaluate the temporal parameters of DNA damage persistence and repair.

First, we investigated whether DNA lesions induced at the S phase could persist until the G1 phase of the next cell cycle. To do this, we studied the effect of CPT on the efficiency of illegitimate hybridization in the strains bearing the *rad52* mutation. CPT causes DNA breaks primarily at the S phase of the cell cycle, which can only be fixed by the Rad52-dependent recombination repair at late S/G2 phases [56,62]. Yeast cells can only hybridize at the G1 phase, which means that the mating type switch leading to the appearance of illegitimate hybrids also manifests at the G1 phase (Figure 6). We hypothesized that if some CPT-induced DNA breaks persisted until the G1 phase of the next cell cycle, this would considerably increase the efficiency of illegitimate hybridization in *rad52* mutants compared to the wild-type strain.

We found that 7 µg/mL of CPT, which was effective in inducing illegitimate hybridization in the wild-type strain (Figure 4), was highly toxic to the *rad52* mutant (Figure 7A). This indicates that DNA lesions caused by CPT are lethal in the absence of recombination repair. We then tested whether CPT would stimulate illegitimate hybridization in the *rad52* strains. Since it was not possible to accurately measure the frequency of illegitimate hybridization due to the lethality of CPT for the *rad52* strain, we performed a qualitative test (Figure 7B). We mixed the tester strain (*wt* or *rad52*) and the strain partner for hybridization on YPD media containing CPT in different concentrations. On the next day, plates were replica-plated on media for illegitimate hybrid selection. Under these conditions, even cells with lethal chromosome aberrations caused by CPT could survive due to mating with the partner strain because, as we know now, even the loss of the whole chromosome III does not prevent mating (Figure 3). The amount of spontaneous illegitimate hybrids was higher in the *rad52* mutant than in the wild-type strain (Figure 7B). However, treatment with CPT did not stimulate illegitimate hybridization in *rad52* mutants, while it significantly increased the number of illegitimate hybrids in the wild-type strain (Figure 7B,C). A few colonies of illegitimate hybrids, appearing on selective media after CPT treatment of the *rad52* strain (Figure 7B), may have emerged as a result of spontaneous genetic events, potentially occurring even before spreading them on the CPT-containing medium.

Therefore, we concluded that CPT-induced DNA breaks did not persist from the S phase until the G1 phase of the next cell cycle. Cells bearing the *rad52* mutation do not survive on CPT-containing media, while in the wild-type cells, CPT-induced primary DNA lesions can turn into inheritable changes in genetic material through recombination repair, leading to mating-type switching, and can be detected in the alpha-test (Figure 3). Also, at least some CPT-induced DSBs could occur at the G1 phase when CPT blocks the topoisomerase I involved in transcription [56].

We then asked whether primary DNA lesions occurring specifically in the G1 phase would be more effectively detected in the alpha-test, as lesions occurring in this phase have the greatest potential to affect the mating type. To answer this question, we conducted the illegitimate hybridization and cytoduction tests using asynchronous cultures of the *wt* strain and cultures of a temperature-sensitive mutant strain *cdc28-4*. The *cdc28-4* allele encodes for a protein with a single amino acid substitution H128Y, leading to a START-defective phenotype [63,64,65]. Mutant strain *cdc28-4* undergoes arrest at the G1 phase at 37 °C due to the disruption in the function of the cyclin-dependent kinase Cdc28 [66]. In this experiment, primary DNA lesions were induced by UV irradiation, as UV showed high efficiency in the induction of mating-type switching (Figure 4).

Cells bearing the *cdc28-4* mutation grew more slowly compared to *wt* cells and had a twice as long doubling time, even at permissive temperature (20–22 °C) [67]. This suggests that the *cdc28-4* strain has a partial cell cycle arrest at 22 °C. Under these conditions, the frequency of spontaneous illegitimate hybridization and cytoduction in mutant cells was significantly higher than in *wt* cells, due to an increase in the frequencies of all scored genetic events (hybridization *wt* 375 × 10^−7^ and *cdc28-4* 611 × 10^−7^ without UV; cytoduction *wt* 5.5 × 10^−7^ and *cdc28-4* 24 × 10^−7^ without UV). Furthermore, mutant cells arrested in the G1 phase at the restrictive temperature of 37 °C showed a more substantial increase in the relative frequencies of UV-induced illegitimate hybridization (16-fold), especially in mutations and temporary lesions (49-fold). We further separately analyzed the frequency of primary lesions and all inheritable changes in genetic material. We compared the frequency of illegitimate cytoductants with α mating type (primary lesions) to the frequency of cytoductants of mating type a, recessive a, and non-maters (heritable changes). Our data indicate that at 22 °C, the frequency of induced primary DNA lesions in the *cdc28-4* mutant was 2.5 times higher than in the *wt* cells, while at 37 °C, the difference between these parameters was 12-fold (Figure 8A). As the results of the illegitimate cytoduction test show, in cultures arrested at the G1 phase, 99% of illegitimate cytoductants exhibited the alpha mating type, indicating that they result from a temporary mating type switch (Figure 8B). In contrast, in the asynchronous culture of the wild-type strain, the ratio of alpha to non-alpha cytoductants was 88:12% (Figure 8B). Therefore, the prolonged G1 phase of the cell cycle allows more primary lesions to express phenotypically and induce the mating type switch. After mating, some of these lesions are removed by repair, and the others are fixed as mutations.

We have shown here that the alpha-test is a useful tool to study the timing of primary lesions’ persistence in the cell cycle. This is possible not only because the alpha-test allows one to detect the phenotypic expression of primary DNA lesions before they are eliminated by repair, but also due to the close connection between the genetic control of mating type and the cell cycle. The mating of two yeast cells is possible only when both coupling cells are synchronized in the G1 phase. So, in the alpha-test it is guaranteed that the expression of a primary lesion took its place at the G1 of the cell cycle. We have shown that heterothallic yeast cells do not need to undergo a complete cell cycle to get rid of products specific for the α-mating type cells. If a primary DNA lesion occurs in the G1 phase of the cell cycle, it can immediately lead to a mating type switch α → a in a yeast cell and express phenotypically during the same stage of the cell cycle. We assume that if primary lesions arise in other phases of the cell cycle (S, G2, or M) they are likely eliminated by repair before the next G1 when yeast cells are competent for mating. Most of those primary lesions are converted into mutations or eliminated in an error-free manner. In this case, the proportion of phenotypically expressed primary lesions is significantly smaller, and most of them will be detected as resulting endpoints.

The phenotypic expression of primary lesions that occurred in S or G2 would depend on their type and ability to block replication. For example, base modifications induced by some base analogs do not block replication or transcription and potentially may exist in the cell during several cell cycles before they are converted into mutations [68,69]. DSBs must be repaired by homologous recombination at the S/G2 phases of the cell cycle, otherwise, the cells will not survive. DSBs cannot exist in a cell longer than one cell division. Thus, DSBs that occur outside the G1 phase do not have phenotypic expression and will be detected in the alpha-test as inheritable genetic changes. DSBs can be phenotypically expressed only if they arise in the G1 phase when mating is possible.

## 3. Materials and Methods

### 3.1. Yeast Strains. S. cerevisiae Strain

K5-35B-D924 (*MATα ura3Δ leu2Δ met15Δ lys5::KanMX cyh^r^*) was used for the illegitimate hybridization test (Table 4). Isogenic to K5-35B-D924 *kar1-1* mutant without mitochondrial DNA ([*rho^0^*]) was used for illegitimate cytoductant selection because of its increased cytoduction frequency. Since *kar1-1* reduces the frequency of hybridization, different isogenic strains were used in hybridization and cytoduction tests. For cytoductant selection, nuclear recessive mutation *cyh^r^* (cycloheximide resistance) and cytoplasmic marker [*rho^0^*] were used. Diploid strain D926 homozygous for *MATα* (*MATα//MATα ADE1//ade1Δ leu2Δ//leu2Δ lys2Δ//lys2Δ ura3Δ//ura3Δ his4Δ//his4Δ thr4Δ//thr4Δ CYH^s^ [rho^+^]*) was used as a partner for mating. The diploid status of this strain was required to reduce the influence of genome instability in the mating partner on the frequency of illegitimate hybridization and cytoduction. Recessive mutations *his4Δ* and *thr4Δ* in the D926 strain were used to mark both arms of chromosome III. On the base of the K5-35B-D924 strain and its *kar1-1* [*rho^0^*] derivative, *cdc28-4* was constructed as described in [67], the *ogg1Δ::URA3* deletion was created by using plasmid pPM867 [70], and *pms1Δ::LEU2* mutants were introduced using plasmid PAM58 [71]. Strain K5-35B-D924-cen3 carrying insertion of the *GAL1* promoter in the centromere region of chromosome III was obtained by transforming the K5-35B-D924 and K5-35B-D924 *kar1-1* strains with the plasmid pCEN-UG (kindly provided by Prof. Reid, Columbia University Medical Center, USA) linearized by restriction with NotI [55]. Strains 78A-P2345 (*MAT*α *his5*), 2G-P2345 (*MATa*
*his5*), and 2A-P143 (*MAT*α *ura3Δ ade8Δ*) were used as mating type testers. Strains defective in the recombination repair pathway were constructed by disruption *rad52::LEU2*. To do this, the yeast strains K5-35B-D924 and K5-35B-D924-kar1 were transformed with the plasmid pJH181 [72,73]. The plasmid was pre-treated with the restriction endonuclease BamHI before transformation. Transformants were selected on synthetic media without leucine.

### 3.2. Media and Growth Conditions

Cells were grown on either rich YPD media (1% yeast extract, 2% peptone, 2% dextrose) or minimal synthetic media (minimal dextrose, MD) containing yeast nitrogen base with ammonium sulfate (0.67%), glucose (2%), agar for solid media (2%) and specific amino acids and nitrogenous bases required for selection. All liquid and solid media were prepared according to standard protocols [74]. Minimal medium lacking glucose and containing ethanol (20 mL/L), cycloheximide (5 µg/mL), and all nutrients required for K5-35B-D924 *kar1-1* strain growth, was used for cytoductants selection. Yeast cultures were grown at 30 °C or 22 °C. Cells were incubated at 37 °C for 2 h to block *cdc28-4* mutants at the G1 phase of the cell cycle.

### 3.3. Illegitimate Mating Assay (Alpha-Test)

Six to twelve independent fresh overnight cultures of K5-35B-D924 or K5-35B-D924-kar1-1 were used for illegitimate hybridization or cytoduction. For the illegitimate hybridization assay, 50–100 μL of the K5-35B-D924 (tester strains) or its derivative culture was plated on the selective minimal media containing histidine, threonine, leucine, and uracil, then 100 μL of D926 culture was added to each plate and the mix of both strains was evenly spread on the agar surface. In parallel, the same culture of the tester strain was diluted to an appropriate cell density and plated on the YPD media for survival measurement. The dilution factor varied from approximately 20,000 to 40,000 depending on the survival rate of the strain, which allowed us to obtain a convenient number of colonies for counting on each plate (from 100 to 500 cfu/plate). Both surviving colonies and hybrids were scored after 3 days of incubation (30 °C) to calculate the frequency of the illegitimate hybridization. For the illegitimate cytoduction assay, cultures of K5-35B-D924-kar1-1 and D926 strains were concentrated up to ~10^9^ cells/mL. An amount of 50–100 μL of tester strain and the mating partner cells suspension were mixed on the YPD agar plate, incubated for 2 days at 30 °C, and then replica plated onto the selective media for cytoductants selection. In parallel, the same cultures of the tester strain were diluted to appropriate cell densities and plated on YPD for survival measurement. Surviving colonies were scored after 2 days, and cytoductants were scored after 10 days of incubation. To identify what kind of genetic change induced the mating type switch, we determined the phenotype of illegitimate hybrids and cytoductants. For each experimental condition, 500–2000 colonies of illegitimate hybrids or cytoductants were randomly picked up and plated on a master plate with short patches. Then, the master plate was replica plated on dropout media, and the mating type of illegitimate hybrids and cytoductants was revealed by hybridization with testers. To determine whether the phenotype of “a” mating type cytoductants is recessive, we crossed them with strain 2A-P143, and then checked the mating type of the resulting hybrids. If the initial cytoductant had a recessive mating type “a”, then the hybrid derived from it had an “α” mating type. The sterility of the hybrid indicated that the original cytoductant had semi-dominant mating type “a” and originated due to conversion between *HMRa* and *MATα*. Then, hybrids and cytoductants were classified based on their phenotype according to Table 1 and Figure 2. The ratio and frequency of each class of hybrids or cytoductants were determined (Table 2 and Table 3).

### 3.4. Treatment with Mutagens and Disruption of DNA Repair Systems

Yeast cells were treated with camptothecin (CPT) and UV radiation. CPT treatment was conducted as follows: the overnight culture of the tester strain together with mating partner D926 was plated on YPD medium containing CPT (7 µg/mL) and on CPT-free media. Both strains were incubated together on a complete agar media overnight and then replica plated onto the media for illegitimate hybrid or cytoductant selection. For UV treatment, the tester strains were plated on solid complete and selective media and then irradiated with UV light (20 J/m^2^, λ = 254 nm). The D926 strain was added after irradiation. The *ogg1* mutation was used to disrupt the base excision repair. The *pms1* mutation was used to impair the mismatch bases repair. Tester strains with mutations were incubated overnight and plated together with mating partner D926 on selective media.

### 3.5. Assessment of DNA Lesion Stability in the Cell Cycle (in the G1 Phase)

To track primary DNA lesions during the G1 phase of the cell cycle, we determined the ratio of inherited and non-inherited genetic changes induced by UV radiation in asynchronous yeast cultures and cultures blocked at the G1 phase. To prevent the transition of yeast cells from the G1 to the S phase, we used temperature-sensitive mutation *cdc28-4*, disrupting the activity of the cyclin-dependent kinase Cdc28 at 37 °C. The frequencies of illegitimate hybridization and cytoduction were determined as described above with the following modifications. For the experiment, we used 9 independent cultures of each strain (wild-type and *cdc28-4*) grown in liquid media at 22 °C for 48 h. Then, aliquots of each culture of wild-type and *cdc28-4* strains were plated on four plates containing media selective for illegitimate hybrids and incubated for 2 h at 37 °C. Then, two of the four plates were treated with UV; control variants were not irradiated. After the UV treatment, 100 µL of overnight D926 culture was spread on each plate. Then, the UV-irradiated plates and nontreated control plates were incubated either at 22 °C or at 37 °C (*cdc28-4* was permanently blocked at the G1 phase). For the survival measurement, appropriate dilutions of each culture were plated on complete media, then UV-irradiated and incubated under the same conditions. The dilution factor varied from approximately 20,000 to 40,000 depending on the survival rate of the strain, which allowed us to obtain a convenient number of colonies for counting on each plate (from 100 to 500 cfu/plate).

### 3.6. Statistical Analysis

The Mann–Whitney or Kruskal–Wallis test (independent samples) and Wilcoxon or Friedmann test (dependent samples) were used to determine statistical significance for the frequencies of spontaneous and induced illegitimate hybridization/cytoduction. Frequencies (*f*) are presented as medians and 95% confidence intervals [75].

The frequency of each class of genetic events detected in the illegitimate hybridization and cytoduction test was calculated by multiplying the proportion of the corresponding event class by the overall frequency of hybrids or cytoductants. The significance of the differences between the proportions of hybrid or cytoductant classes was evaluated by the Z-test for proportions with Yates continuity correction and Bonferroni correction for multiple comparisons [76]. Confidential intervals for proportions were calculated by the Exact (Clopper–Pearson) method or Score (Wilson) method [77].

## 4. Conclusions

We have concluded that the phenotypic expression of a primary DNA lesion in the alpha-test depends on the type of lesion, as different lesions have varying abilities to disrupt the chemical structure of DNA and therefore affect transcription and replication. Additionally, the phase of the cell cycle in which the lesion occurs is also important due to the specificity of the test system used, which allows for the detection of changes in phenotype only at the G1 stage. When applying our findings to other organisms, it is important to consider if the expression of a lesion may depend on the specific stages of the cell or life cycle, as well as the tissue type where the corresponding gene is expressed, and environmental factors involved in the regulation of the gene expression. Lesions in different genes may have different impacts on the phenotype. Our data contribute to understanding the mechanisms of hereditary and modification variability. The phenotypic expression of primary DNA lesions in the *MATα* locus in yeast that are repaired without errors after mating is an example of modification changes. It is possible to find examples of modification changes in other species: morphs in *Drosophila melanogaster* or congenital but non-inherited changes in mammals, which can occur as a result of the phenotypic expression of primary lesions during certain periods of organism development. Thus, hereditary and modification changes may have a common origin and molecular mechanism: the error-prone or error-free repair of primary lesions. It can be assumed that primary lesions can persist for a long time in dividing cells, accumulate, and disrupt gene expression, thereby affecting cell phenotype. If such a lesion occurs in repair genes, it may lead to a transient increase in the level of mutagenesis, the accumulation of mutations, and, subsequently, an increase in the level of hereditary variability.

## Figures and Tables

**Figure 1 ijms-24-12163-f001:**
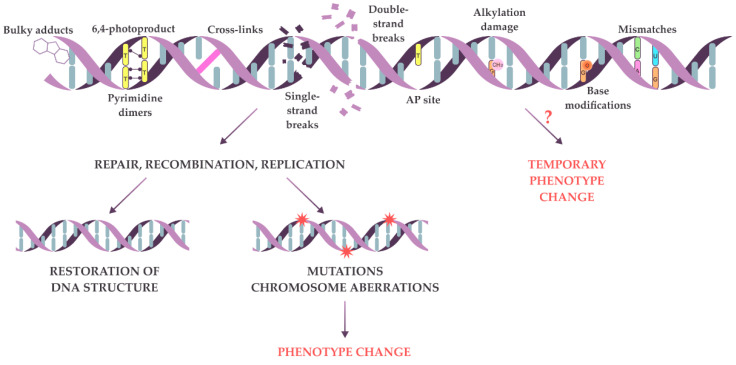
Primary DNA lesions and possible consequences of repair on damaged DNA. Common types of DNA primary lesions are shown at the top. The damaged DNA can be repaired (bottom left) with the restoration of the initial sequence or the introduction of mutations (red stars) leading to phenotypic changes. Alternatively, primary lesions may cause temporary phenotypic changes on their own (bottom right). However, this possibility has not been extensively tested and remains hypothetical for many types of DNA lesions (marked by “?”).

**Figure 2 ijms-24-12163-f002:**
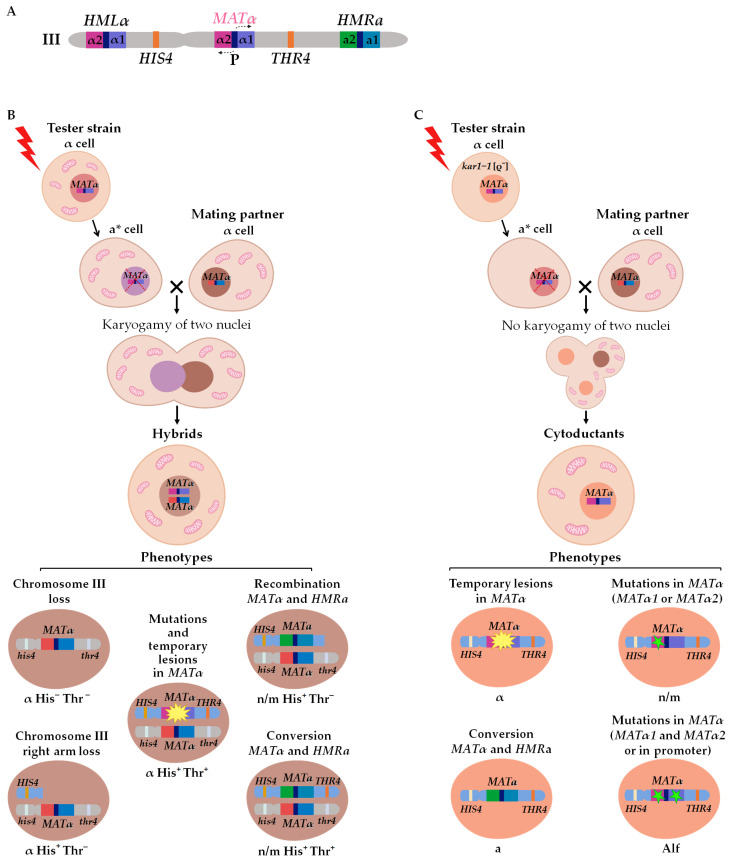
The alpha-test: scheme of chromosome III (**A**), genetic changes leading to illegitimate hybridization, (**B**) or cytoduction (**C**). a*—temporary “a” mating type.

**Figure 3 ijms-24-12163-f003:**
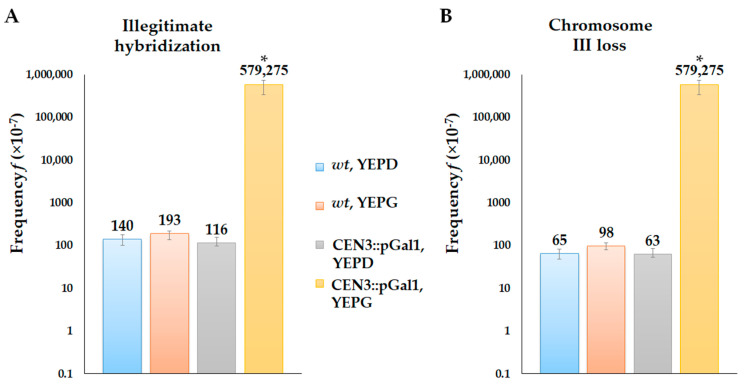
Chromosome III loss induced by the transcription in the centromere sequences detected by illegitimate hybridization. The frequency of illegitimate hybridization (**A**) and the frequency of chromosome III loss (**B**) in the *wt* and *CEN3::pGal1* strains on glucose and galactose media. *—Statistically significant difference between *CEN3::pGal1* strain on galactose and wild-type strain grown on glucose and galactose (Mann–Whitney test) and *CEN3::pGal1* strain on glucose (Wilcoxon test).

**Figure 4 ijms-24-12163-f004:**
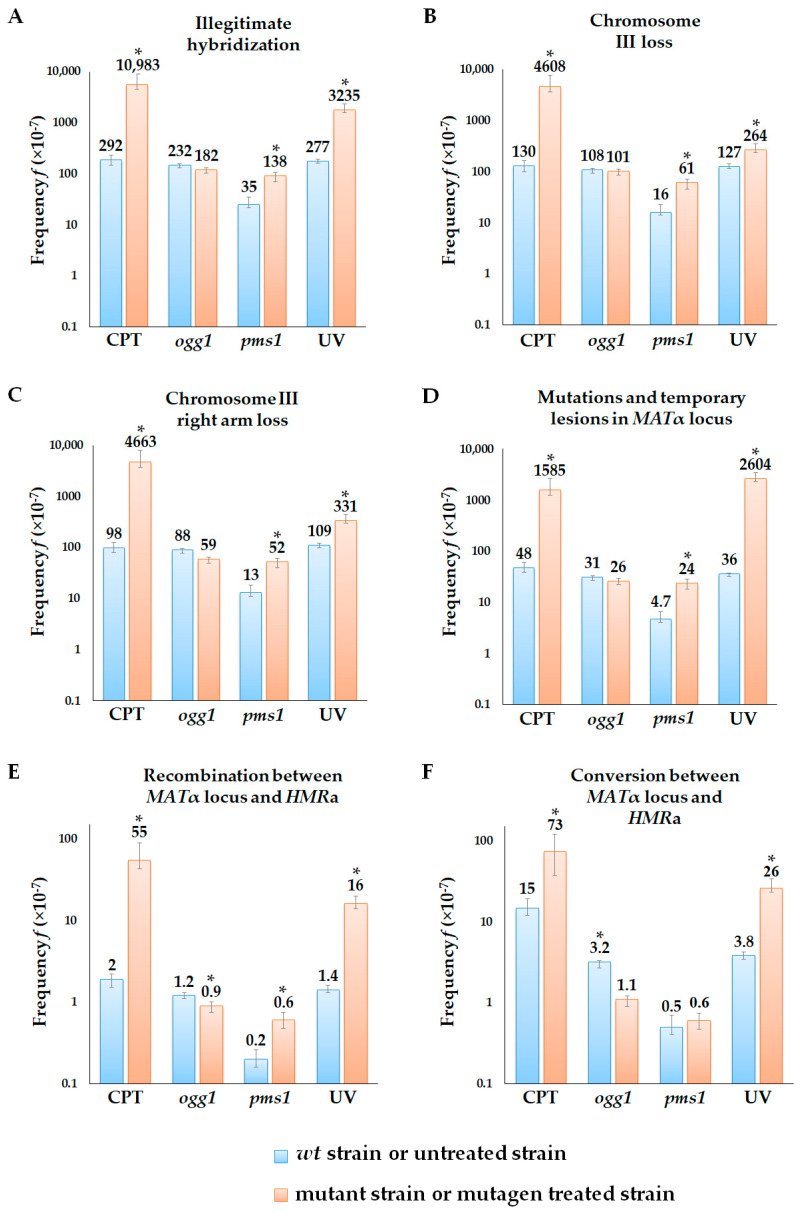
Frequency of genetic changes induced by reference mutagens and repair system defects detected in the illegitimate hybridization test: (**A**)—illegitimate hybridization; (**B**)—chromosome III loss; (**C**)—chromosome III right arm loss; (**D**)—mutations and temporary lesions in *MATα* locus; (**E**)—recombination between *MATα* locus and *HMRa*; (**F**)—conversion between *MATα* locus and *HMRa*. *—Statistically significant difference between the treated or mutant strain and control variants, estimated by Mann–Whitney test for independent samples or Wilcoxon signed-rank test for dependent samples.

**Figure 5 ijms-24-12163-f005:**
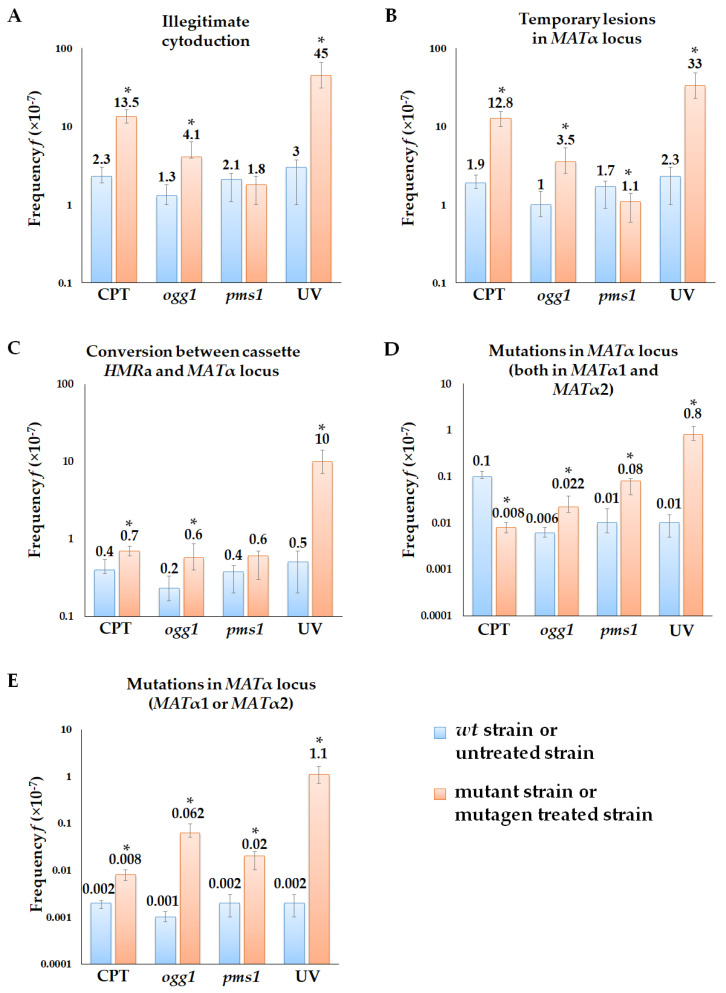
Frequency of genetic changes induced by reference mutagens and repair system defects detected in the illegitimate cytoduction test: (**A**)—illegitimate cytoduction; (**B**)—temporary lesions in the *MAT*α locus (both in *MAT*α*1* and *MAT*α*2* or in the bidirectional promoter); (**C**)—gene conversion between cassette *HMRa* and *MATα* locus; (**D**)—mutations in the *MAT*α locus (both in *MAT*α1 and *MAT*α2 or in the bidirectional promoter); (**E**)—mutations in the *MAT*α locus (*MAT*α1 or *MAT*α2). *—Statistically significant difference between the treated or mutant strain and control variants, estimated in the Mann–Whitney test for independent samples or Wilcoxon signed-rank test for dependent samples.

**Figure 6 ijms-24-12163-f006:**
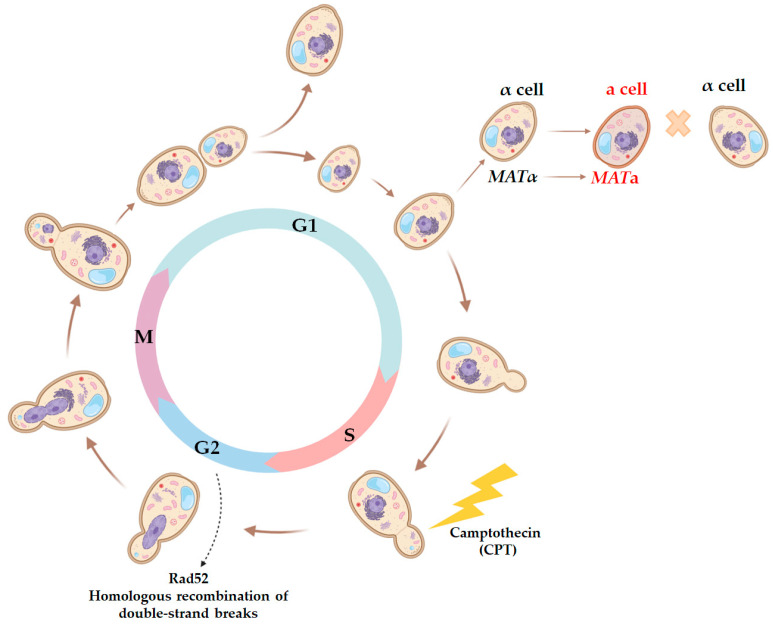
Scheme of the yeast cell cycle representing physiological markers for each phase and critical points when yeast cells are capable of mating or the most sensitive to certain DNA-damaging agents. Since hybridization in yeast is only possible when both cells are synchronized at the G1 stage, the use of mutagenic factors acting at specific stages of the cell cycle allows for the assessment of the ability of a primary lesion to persist over several stages of the cell cycle. Thus, CPT predominantly causes DNA breaks at the S stage, and then they may be repaired by the homologous recombination system in an error-free or error-prone manner at the G2 stage. The repair result can be accounted for at the G1 stage by changes in the frequency of illegitimate hybridization and the phenotype of illegitimate hybrids.

**Figure 7 ijms-24-12163-f007:**
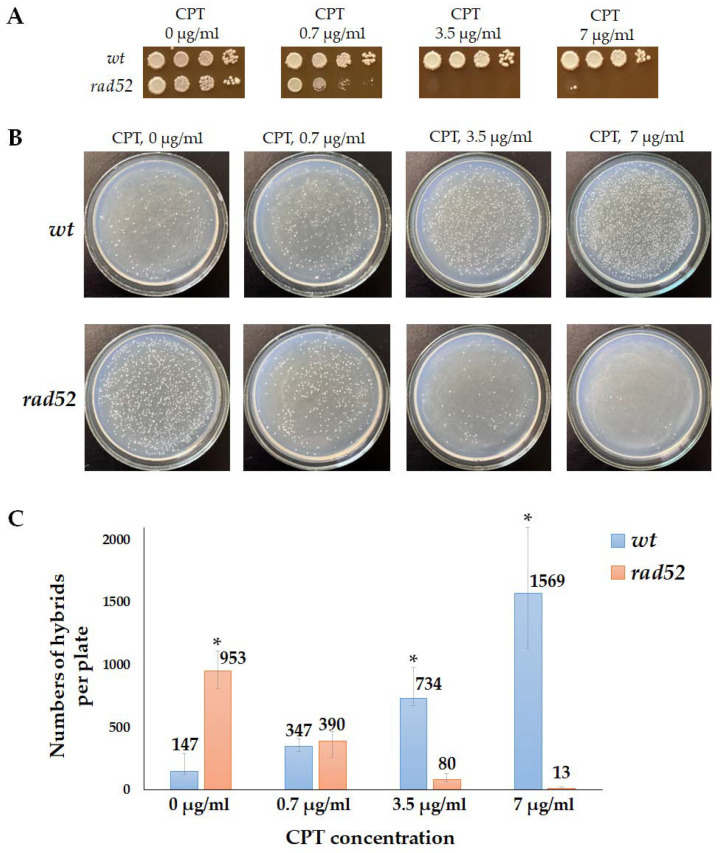
Effect of camptothecin (CPT) on the efficiency of illegitimate hybridization in *wt* and *rad52* strains: (**A**)—*wt* and *rad52* sensitivity to CPT. Ten-fold serial dilutions of the *wt* and *rad52* overnight cultures were spotted on YPD media, containing CPT in different concentrations, and incubated at 30 °C for 3 days. (**B**)—Qualitative illegitimate hybridization test. The tester strain (*wt* or *rad52*) and the strain partner for hybridization (D926) were plated together on YPD media containing CPT in different concentrations. The next day the plates were replica-plated on media for illegitimate hybrid selection and incubated at 30 °C for 3 days. (**C**)—Number of hybrids per plate (median and CI). *—Statistically significant difference between the *wt* and *rad52* strains estimated by the Mann–Whitney test for independent samples.

**Figure 8 ijms-24-12163-f008:**
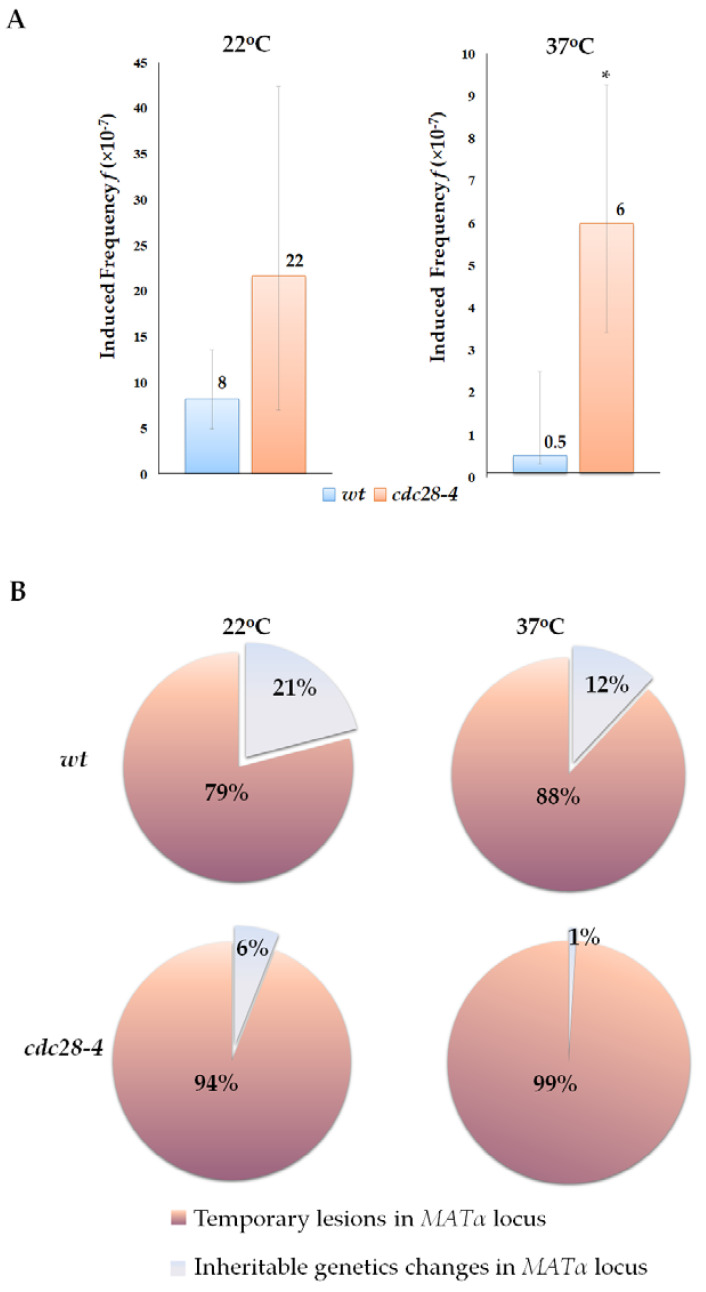
Effect of G1 phase prolongation on the phenotypic expression of primary DNA lesions in illegitimate cytoduction. (**A**)—UV-induced frequency of primary DNA lesions in *wt* and *cdc28-4* strains grown at 22 °C and 37 °C; *—statistically significant difference in UV-induced frequency between the *wt* and *cdc28-4* strains estimated by the Mann–Whitney test for independent samples. (**B**)—The ratio of inheritable and transient changes in the genetic material in *wt* and *cdc28-4* strains grown at 22 °C and 37 °C.

**Table 1 ijms-24-12163-t001:** Genetic events leading to the illegitimate mating of *MATα his4 thr4* strain to *MATα HIS4 THR4* and the phenotypes of the resulting hybrids and cytoductants.

**Genetic Event**	**Hybrid Phenotype**	**Cytoductant Phenotype**
Conversion between *HMRa* and *MATα* locus	n/m His^+^Thr^+^	a
Reciprocal recombination between *MATα* locus and *HMRa*	n/m His^+^Thr^−^	lethal
Loss of the right arm of chromosome *III*	α His^+^Thr^−^	
Loss of chromosome *III*	α His^−^Thr^−^	
Mutations in *MATα* (*matα1* or *matα2*)	α His^+^Thr^+^	n/m
Mutations in *MATα* (both in *MAT*α*1* and *MAT*α*2* or in the bidirectional promoter, *MATα* deletions)		recessive “a” mating type or Alf-phenotype (from a-like fakers)
Temporary lesions in the *MAT*α locus (both in *MAT*α*1* and *MAT*α*2* or in the bidirectional promoter)		α

n/m—non-mating phenotype.

**Table 2 ijms-24-12163-t002:** Frequencies and proportions of spontaneous genetic events scored in the illegitimate hybridization assay in *S. cerevisiae* strain K5-35B-D924.

Genetic Event	Frequency × 10^−7^ (Median and Conf. Int.)	Proportion of Genetic Event, % (Conf. Int.)
Illegitimate hybridization (all classes)	_83_ 206 _386_	100
Classes:		
Chromosome III loss	_37_ 96 _180_	_44_ 46.6 _49_
Loss of chromosome III right arm	_34_ 78 _147_	_35_ 38 _40_
Mutations or temporary lesions in *MATα*	_11_ 27 _52_	_12_ 13.5 _15_
Reciprocal recombination between *MATα* and *HMRa*	_0.4_ 1.1 _2_	_0.2_ 0.5 _1_
Conversion *HMRa* → *MATα*	_1.1_ 2.8 _5.2_	_0.8_ 1.4 _2.1_

**Table 3 ijms-24-12163-t003:** Frequencies and proportions of spontaneous genetic events scored in the illegitimate cytoduction assay in *S. cerevisiae* strain K5-35B-D924-kar1-1 [*rho^−^*].

Genetic Event	Frequency × 10^−7^, (Median and Conf. Int.)	Proportion of Genetic Event, % (Conf. Int.)
Illegitimate cytoduction (all classes)	_1.7_ 2.1 _2.4_	100
Classes:		
Temporary lesions in *MATα*	_1.4_ 1.7 _1.9_	_79_ 81.1 _83_
Conversion *HMRa* → *MATα*	_0.31_ 0.38 _0.43_	_16_ 18.3 _20_
Mutations in *MAT*α (both in *MAT*α*1* and *MAT*α*2* or in the bidirectional promoter, *MATα* deletions)	_0.008_ 0.009 _0.01_	_0.2_ 0.5 _1_
Mutations in *MATα* (*matα1* or *matα2*)	_0.001_ 0.0016 _0.002_	_0.01_ 0.1 _0.4_

**Table 4 ijms-24-12163-t004:** *Saccharomyces cerevisiae* strains used in this study.

Strain	Genotype	Application
K5-35B-D924	*MATα ura3Δ leu2Δ met15Δ lys5::KanMX cyh^r^*	Tester strain for illegitimate hybridization
K5-35B-D924-ogg1	*MATα ura3Δ leu2Δ met15Δ lys5::KanMX cyh^r^ ogg1Δ::* *URA3*	
K5-35B-D924- pms1	*MATα ura3Δ leu2Δ met15Δ lys5::KanMX cyh^r^ * *pms1Δ::LEU2*	
K5-35B-D924-cdc28-4	*MATα ura3Δ leu2Δ met15Δ lys5::KanMX cyh^r^ cdc28-4*	
K5-35B-D924-rad52	*MATα* *ura3Δ* *leu2Δ* *met15Δ* *lys5::KanMX cyh^r^ rad52::LEU2*	
K5-35B-D924-kar1-1	*MATα ura3Δ leu2Δ met15Δ lys5::KanMX cyh^r^ kar1-1 [rho^−^]*	Tester strain for illegitimate cytoduction
K5-35B-D924-ogg1-kar1-1	*MATα ura3Δ leu2Δ met15Δ lys5::KanMX cyh^r^ ogg1Δ::* *URA3 kar1-1 [rho^−^]*	
K5-35B-D924-pms1-kar1-1	*MATα ura3Δ leu2Δ met15Δ lys5::KanMX cyh^r^* *pms1Δ::LEU2 kar1-1 [rho^−^]*	
K5-35B-D924-cdc28-4-kar1-1	*MATα ura3Δ leu2Δ met15Δ lys5::KanMX cyh^r^* *cdc28-4 kar1-1 [rho^−^]*	
D926	*MATα//MATα ADE1//ade1Δ leu2* *Δ//leu2* *Δ lys2* *Δ//lys2* *Δ ura3* *Δ//ura3* *Δ his4* *Δ//his4* *Δ thr4* *Δ//thr4* *Δ CYH^s^ [rho^+^]*	Partner for mating
78A-P2345	*MAT*α *his5*	Testers for mating type determination
2G-P2345	*MATa* *his5*	
2A-P143	*MAT*α *ura3Δ ade8Δ*	Tester for identification of cytoductants having Alf phenotype

## Data Availability

Non-applicable.

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
