# Peer review of "Detection of Primary DNA Lesions by Transient Changes in Mating Behavior in Yeast *Saccharomyces cerevisiae* Using the Alpha-Test"

_ijms, 2023, doi:10.3390/ijms241512163_

Round 1

Reviewer 1 Report

1. Please, add an explanation on intoduction part why rad52 strain used in this experiment?

2. Figure 7a, please add the information how many inoculum cells used in the spot tests on both wt and rad52 strain. 

Author Response

Reply to Reviewer 1

The reviews are reproduced in blue, our responses to each point raised are below in black. We hope that the revised manuscript meets your expectations.

A: We appreciate the reviewer's suggestions and have now included changes improving the clarity of the manuscript.

R: 1. Please, add an explanation on introduction part why rad52 strain used in this experiment?

A: We thank reviewer for the suggestion. However, we feel that providing an explanation for the rationale of using the rad52 strain in the Introduction part would result in unnecessary duplication, because it is provided at the beginning of section 2.3.

R: 2. Figure 7a, please add the information how many inoculum cells used in the spot tests on both wt and rad52 strain.

A: To answer the comment, we made changes to legend of Fig 7:

Figure 7. Effect of camptothecin (CPT) on efficiency of illegitimate hybridization in wt and rad52 strains. A – wt and rad52 sensitivity to CPT. Ten-fold serial dilutions of the wt and rad52 overnight cultures were spotted on YPD medium, containing CPT in different concentrations, and incubated at 30°C for 3 days. B – qualitative illegitimate hybridization test. Tester strain (wt or rad52) and the strain partner for hybridization (D926) were plated together on YPD media containing CPT in different concentrations. The next day the plates were replica-plated on media for illegitimate hybrids selection and incubated at 30°C for 3 days. C. Number of hybrids per plates (median and CI). * - statistically significant difference between the wt and rad52 strains estimated in Mann–Whitney test for independent samples.

Reviewer 2 Report

The manuscript presents a very impressive body of work connected to phenotypic effects of primary DNA lesions. The scientific value, soundness and novelty of the data is undisputable. The presentation of the results is overall quite logical, but I do have some questions and suggestions.

Abstract: I would suggest slightly shifting the focus of the abstract to more precise description of the obtained results and conclusions. From the first reading of the abstract (before reading the whole manuscript), I found the description of results with “different mutagens causing specific DNA damage” and “repair defects” very unclear. Mentioning the particular mutagens, associated types of DNA lesions and mutated genes right in the abstract would make the impression much clearer. Besides, the results about the conditionally stable chromosome III have important implications even outside the alpha-test and related techniques and should definitely make their way to the abstract.

Fig 1: The design is very nice, but the letters designating the nucleotides and radicals are extremely hard to read. It would be great to make them larger. In addition, using the beads somewhere inside the DNA helix as a symbol for bulky adducts remained unclear to me. Maybe the problem was on my side, but then I would be grateful for an explanation.

Fig. 2: the text is also very small (for example, when compared to the font size in the main text), and it would be nice to make it larger (and legible at 100% zoom).

L88-90 “To the best of our knowledge, the alpha-test is the only system that identifies different types of changes in DNA, point mutations, recombination, whole chromosome or chromosome arm loss, and primary lesions”

I don’t argue with the authors at all, but it would be a really nice place to explain the similarities and differences between the alpha-test and A-like faker assay (for example, https://link.springer.com/protocol/10.1007/978-1-4939-7306-4_1), because they can easily be confused by an not so experienced reader.

L154 “Cytoduction is highly stimulated by the kar1-1 mutation [48-50]”: is it possible to say a few words about the function of the Kar1 protein and the changes triggered by kar1-1? Moreover, is there an English translation of this manuscript [50], and if yes, would it make sense to cite it instead?

Table 3. Why is the frequency of temporary lesions in MATα (1.5x10^7) corresponding to 81 % higher than the total frequency of illegitimate cytoductants (10^7 and 100%)? I thought that the classes are determined after the selection of cytoductants. If I am mistaken, please explain in more detail how that could happen.

L222-223 “We inserted an inducible pGAL1 promoter adjacent to the centromere DNA and transferred the cells onto a galactose medium [51].”: the reference at the end of this sentence looks a bit strange...

L245 “less severe DNA lesions”: is there a definition of more / less severe or a classification?

L452 and elsewhere: are the nature of the cdc28-4 mutation and its particular effect on the protein function known?

L462: “pJH181”: I failed to find some explanation or link to this plasmid...

L469: “ethylanol”: is it a typo for “ethanol” or something else?

L523: “appropriate dilutions”: were these dilutions 5- or 10-fold and how many cells were planted, for example, in the first dilution? It would be nice to add this information for reproducibility and to make comparison with other strains possible.

The language is absolutely clear and understandable, but the manuscript could benefit from a round of proofreading before the final publication. Please see some suggestions below (definitely not the full list!).

L27-28 “DNA lesions by themselves may also have transcriptional effects” => “DNA lesions may have their own transcriptional effects”

L33-34 “Using the alpha-test and different mutagens causing specific DNA damage in combination

with repair defects, we found which primary lesions affected the phenotype.” unclear...

is “DNA mispair” (L46) an accepted term? I can imagine “mispairing” as a process and “mismatch” as a result, but mispair sounds strange

L53-54 “DNA sequence gets restored without any changes.” => “the initial DNA sequence is restored” ?

L129-130

“HO endonuclease which is necessary for the initiation of conversion process” => “HO endonuclease, which is necessary for the initiation of the conversion process”

L165: “error-freeway” => “error-free way”?

L232-333 “the possibility of hybridization remains preserved” => “the possibility of hybridization remains open”

L359 “As in the Figure 7B,” can be just removed, as it is already mentioned in the end of the sentence.

L387 “has a partial the cell cycle arrest at 22°С.” => “has a partial cell cycle arrest at 22°С.”

L499: “night culture of the tester strain” (and elsewhere) => “overnight”?

L501: “during G1 phase of the cell cycle”=> “the G1 phase”

Author Response

Reply to Reviewer 2

 The reviews are reproduced in blue, our responses to each point raised are below in black. We hope that the revised manuscript meets your expectations.

R: The manuscript presents a very impressive body of work connected to phenotypic effects of primary DNA lesions. The scientific value, soundness and novelty of the data is undisputable. The presentation of the results is overall quite logical, but I do have some questions and suggestions.

A: We appreciate reviewer’s positive assessment of our work and thoughtful suggestions for improvement of our article.

  1. Abstract: I would suggest slightly shifting the focus of the abstract to more precise description of the obtained results and conclusions. From the first reading of the abstract (before reading the whole manuscript), I found the description of results with “different mutagens causing specific DNA damage” and “repair defects” very unclear. Mentioning the particular mutagens, associated types of DNA lesions and mutated genes right in the abstract would make the impression much clearer. Besides, the results about the conditionally stable chromosome III have important implications even outside the alpha-test and related techniques and should definitely make their way to the abstract.

A: We made changes:

Abstract: Spontaneous or induced DNA lesions can result in stable gene mutations and chromosomal aberrations due to their inaccurate repair, ultimately resulting in phenotype changes. Some DNA lesions per se may interfere with transcription leading to temporary phenocopies of mutations. The direct impact of primary DNA lesions on phenotype before their removal by repair is not well understood. To address this question, we used the alpha-test that allows detecting various genetic events leading to temporary or hereditary changes in mating type α → a in heterothallic strains of yeast Saccharomyces cerevisiae. Here, we compared yeast strains carrying mutations in DNA repair genes: mismatch repair (pms1), base excision repair (ogg1), and homologous recombination repair (rad52), as well as mutagens causing specific DNA lesions (UV light and camptothecin). We found that double-strand breaks and UV-induced lesions have a stronger effect on the phenotype than mismatches and 8-oxoguanine. Moreover, the loss of the entire chromosome III leads to an immediate mating type switch α → a and does not prevent hybridization. We also evaluated the ability of primary DNA lesions to persist through the cell cycle by assessing the frequency of UV-induced inherited and non-inherited genetic changes in asynchronous cultures of a wild-type (wt) strain and in a cdc28-4 mutant arrested in the G1 phase. Our findings suggest that phenotypic manifestation of primary DNA lesions depends on their type and the stage of the cell cycle in which it occurred.

R: 2. Fig 1: The design is very nice, but the letters designating the nucleotides and radicals are extremely hard to read. It would be great to make them larger. In addition, using the beads somewhere inside the DNA helix as a symbol for bulky adducts remained unclear to me. Maybe the problem was on my side, but then I would be grateful for an explanation.

A: We have enlarged the font of nucleotides and changed the designations for DNA bulky adducts. Previously, the beads were used only to schematize the bulky adducts and to show that this is a large change in DNA.

R: 3. Fig. 2: the text is also very small (for example, when compared to the font size in the main text), and it would be nice to make it larger (and legible at 100% zoom).

A: We have enlarged the font in Figure 2 as suggested by the reviewer.

R: 4. L88-90 “To the best of our knowledge, the alpha-test is the only system that identifies different types of changes in DNA, point mutations, recombination, whole chromosome or chromosome arm loss, and primary lesions”

I don’t argue with the authors at all, but it would be a really nice place to explain the similarities and differences between the alpha-test and A-like faker assay (for example, https://link.springer.com/protocol/10.1007/978-1-4939-7306-4_1), because they can easily be confused by an not so experienced reader.

A: We included a brief description of several tests based on illegitimate hybridization.

In the literature, various genotoxicity tests, based on the evaluation of the frequency of yeast mating-type switching have been described, such as alpha-test, A-like faker assay, test for illegitimate hybridization or cytoduction, and nm-test [20,23-25]. They are all based on the same biological principles and represent variants of the same test system, with differences only in which genetic events affecting the mating type locus can be detected by a particular test system. The spectrum of detectable genetic events depends on the genetic markers carried by the tester strains. Depending on the research aims, it may be sufficient to evaluate the overall frequency of illegitimate hybridization using complementary markers allowing for the selection of illegitimate hybrids. When assessing the frequency of chromosomal disruptions, it is necessary to use strains with genetic markers in both arms of chromosome III. Similarly, to distinguish between gene mutations and primary lesions at the MAT locus, it is important to use approaches and corresponding genetic markers that allow analysis of the haploid nucleus of the tester strain.

  1. 5. L154 “Cytoduction is highly stimulated by the kar1-1 mutation [48-50]”: is it possible to say a few words about the function of the Kar1 protein and the changes triggered by kar1-1? Moreover, is there an English translation of this manuscript [50], and if yes, would it make sense to cite it instead?

A: We included a description of the kar1-1 mutation:

The kar1-1 mutant unable to fuse nuclei after mating (karyogamy-defective) was described in 1976 by James Conde and Gerald Fink. The authors showed that in the kar1-1 mutant, the process of hybridization proceeds normally only until the moment of nuclear fusion, after which a multinuclear zygote (heterokaryon) is formed, which can produce cytoductants in subsequent divisions [52]. The kar1-1 mutation (C448T or Pro150Ser) [53] is recessive and does not cause any obvious disruptions in the cell cycle, except for a block in karyogamy due to disruption of duplication of polar spindle bodies and defects in nuclear and cytoplasmic microtubules [53-55]. Presumably, the kar1-1 mutation disrupts only one function of the protein – karyogamy.

R: 6. Table 3. Why is the frequency of temporary lesions in MATα (1.5x10^7) corresponding to 81 % higher than the total frequency of illegitimate cytoductants (10^7 and 100%)? I thought that the classes are determined after the selection of cytoductants. If I am mistaken, please explain in more detail how that could happen.

A: We have made changes to the Table 3.

R: 7. L222-223 “We inserted an inducible pGAL1 promoter adjacent to the centromere DNA and transferred the cells onto a galactose medium [51].”: the reference at the end of this sentence looks a bit strange…

A: We agree that there reference is somewhat misleading. We removed it.

R: 8. L245 “less severe DNA lesions”: is there a definition of more / less severe or a classification?

We suggested that genotoxic factors causing double- and single-strand breaks, adducts, and abasic sites may be more effective inducers of illegitimate hybridization and cytoduction than mutagens causing minor distortions of the DNA structure and not impeding transcription and replication in the MATα locus. We gave the explanation in the text.

R: 9. L452 and elsewhere: are the nature of the cdc28-4 mutation and its particular effect on the protein function known?

A: We added following information to the 2.3 section where the cdc28-4 allele is mentioned for the first time:

The cdc28-4 allele encodes for a protein with a single amino acid substitution H128Y, leading to START defective phenotype [65-67]. Mutant strain cdc28-4 undergoes arrest at the G1 phase at 37°С due to disruption in the function of the cyclin-dependent kinase Cdc28 [68].

 R: 10. L462: “pJH181”: I failed to find some explanation or link to this plasmid.

A: Unfortunatelly, detailed information about the plasmid is missing in literature. We have added the most recent reference of the plasmid use.

R: 11. L469: “ethylanol”: is it a typo for “ethanol” or something else?

A: Corrected.

R: 12. L523: “appropriate dilutions”: were these dilutions 5- or 10-fold and how many cells were planted, for example, in the first dilution? It would be nice to add this information for reproducibility and to make comparison with other strains possible.

A: We have now included this information in the manuscript to provide a better understanding of our methodology:

The dilution factor varied from approximately 20,000 to 40,000 depending on the survival rate of the strain, which allowed us to obtain a convenient number of colonies for counting on each plate (from 100 to 500 cfu/plate).

R: Comments on the Quality of English Language

The language is absolutely clear and understandable, but the manuscript could benefit from a round of proofreading before the final publication. Please see some suggestions below (definitely not the full list!).

13. L27-28 “DNA lesions by themselves may also have transcriptional effects” => “DNA lesions may have their own transcriptional effects”

A: Corrected

R: 14. L33-34 “Using the alpha-test and different mutagens causing specific DNA damage in combination

with repair defects, we found which primary lesions affected the phenotype.” unclear…

A: Corrected

R: 15. is “DNA mispair” (L46) an accepted term? I can imagine “mispairing” as a process and “mismatch” as a result, but mispair sounds strange

A: We changed term “mispair” to “mismatches”.

R: 16. L53-54 “DNA sequence gets restored without any changes.” => “the initial DNA sequence is restored” ?

A: Thank you, the sentence is corrected.

R: 17. L129-130

“HO endonuclease which is necessary for the initiation of conversion process” => “HO endonuclease, which is necessary for the initiation of the conversion process”

A: Corrected.

R: 18. L165: “error-freeway” => “error-free way”?

A: Corrected.

R: 19. L232-333 “the possibility of hybridization remains preserved” => “the possibility of hybridization remains open”

A: Corrected as follows: Thus, our results clearly demonstrate that the loss of the chromosome III does not lead to immediate cell death, and cells without the chromosome III are able to mate..

R: 20. L359 “As in the Figure 7B,” can be just removed, as it is already mentioned in the end of the sentence.

A: Corrected.

R: 21. L387 “has a partial the cell cycle arrest at 22°С.” => “has a partial cell cycle arrest at 22°С.”

A: Corrected.

R: 22. L499: “night culture of the tester strain” (and elsewhere) => “overnight”?

A: Corrected.

R: 23. L501: “during G1 phase of the cell cycle”=> “the G1 phase”

A: Corrected.

Reviewer 3 Report

The manuscript, titled, “Detection of primary DNA lesions by transient changes of mating behavior in yeast Saccharomyces cerevisiae using the alpha-test” by Zhuk et al. is a comprehensive genetic study that is focused on a relatively neglected area of the literature – the transient effects that mutations have on the phenotype of an organism prior to being repaired/stably altered.  They use a clever assay to determine the effects of different mutations on phenotype, using an alpha-test system in S. cerevisiae to capture these events under various conditions, by various mutagens, and they use sound genetic approaches and techniques to unravel the mechanisms underlying their findings.  Overall, this work exhibits sound design, execution, analysis, and is clearly presented and written.

Notes:

Section 2.1 is outstanding.  Clearly written, clearly explained, with a conciseness and focus that is commendable!  Very well done!

Major comments:

Line 374: Figure 7.  While these plates and the figure are excellent, it requires quantification and repetition.  If the authors have performed this assay on multiple biological replicates, then they should score each set of plates (using the dilution schemes and colony, spot formation) and present it within this figure as a graph or table to allow for a more complete understanding of the variability of these results.  I can fully appreciate the effects seen on the plates that are presented, but it requires further verification and quantification all the same.  I would recommend scoring all the data and presenting the relative changes compared to WT.  Please perform appropriate statistical analysis on the entire data.

Line 433: Figure 8.  In part A reformat, calculate, and add error bars, and perform necessary statistics on the WT versus the cdc28 null mutant.

Line 438: please compile the yeast strains into a table for ease of genotype comparison.  Include the specific alleles for all mutations (auxotrophic markers, repair mutants, etc.). 

Line 436: include details of SC/MD media. 

Minor comments:

Line 71: I would suggest adding verbiage to direct the reader to the appropriate section of the figure.  For example, when discussing the damage that accumulates on the DNA refer to the top part of the figure, or label it part A.  Then when discussing the repair/mutagenic repair I would suggest verbiage indicating the bottom left, or label part B.  And when discussing the temporary phenotypic change prior to repair/mutation add verbiage to indicate the bottom right, or label part c.  These are not significant changes, but I think that the more precision that is presented will facilitate readers to the appropriate sections as needed.  Furthermore, the legend should be a bit more descriptive than it currently is written to allow the reader to interpret the sections – and all the various cartoons images used to depict various types of damage that can accumulate.

Line 78: when discussing 8-oxoG and uracil resulting in protein variant production in humans and E. coli, please add a specific example from the works that you cited – from either organism.  (Similar to how you cite the effects on nitrate reductase U. maydis in the previous portion of the paragraph).

Lines 191-192: The border scheme for the rows that comprise table #1 are a bit distracting as presented.  Please reformat in a more aesthetic, clearer manner.

Line 319: Figure 5E – please reformat the numerical labels on the chart so they do not overlap with different bars (e.g. the WT/untreated vs mutant/mutagen bars).

Line 346-347: Figure 6 legend is too brief and lacks appropriate verbiage and description for the figure.  It requires revision for clarity.

Line 408: This reads like a conclusion section.  I would recommend making these final two paragraphs of the results and discussion into a conclusion section and re-numbering the following sections accordingly.  **** after completing the materials and methods I found that there is a Conclusion section!  Move this before the materials and methods (it will most likely be missed by many readers otherwise).

Grammar:

Line 26: missing punctuation – “Spontaneous or induced DNA lesions can due to their inaccurate repair result in stable…” should read: Spontaneous or induced DNA lesions can, due to their inaccurate repair, result in stable…”

Line 184: Spacing issue after ref 29 and the start of the next sentence.

Line 285-286: “pairing with “C” and od non-canonical pairing with with…” – O believe that the word ‘od’ is a typo.  Furthermore the duplicate use of the word ‘with’ appears to be another typo.

Line 387: “the cdc28-4 strain has a partial the cell cycle arrest” – should read has a partial cell cycle arrest.  Delete the word the.

The language only requires minor editing.

Author Response

Reply to Reviewer 3

The reviews are reproduced in blue, our responses to each point raised are below in black. We hope that the revised manuscript meets your expectations.

R: The manuscript, titled, “Detection of primary DNA lesions by transient changes of mating behavior in yeast Saccharomyces cerevisiae using the alpha-test” by Zhuk et al. is a comprehensive genetic study that is focused on a relatively neglected area of the literature – the transient effects that mutations have on the phenotype of an organism prior to being repaired/stably altered.  They use a clever assay to determine the effects of different mutations on phenotype, using an alpha-test system in S. cerevisiae to capture these events under various conditions, by various mutagens, and they use sound genetic approaches and techniques to unravel the mechanisms underlying their findings.  Overall, this work exhibits sound design, execution, analysis, and is clearly presented and written.

Notes:

Section 2.1 is outstanding.  Clearly written, clearly explained, with a conciseness and focus that is commendable!  Very well done!

A: We appreciate reviewer’s positive assessment of our work and thoughtful suggestions for improvement.

R: Major comments:

1. Line 374: Figure 7. While these plates and the figure are excellent, it requires quantification and repetition. If the authors have performed this assay on multiple biological replicates, then they should score each set of plates (using the dilution schemes and colony, spot formation) and present it within this figure as a graph or table to allow for a more complete understanding of the variability of these results.  I can fully appreciate the effects seen on the plates that are presented, but it requires further verification and quantification all the same.  I would recommend scoring all the data and presenting the relative changes compared to WT.  Please perform appropriate statistical analysis on the entire data.

A: We included additional panel C to the Fig 7 with quantitative data of results represented on the panel B.

R: 2. Line 433: Figure 8.  In part A reformat, calculate, and add error bars, and perform necessary statistics on the WT versus the cdc28 null mutant.

A: We added statistical data to the Fig 8.

R: 3. Line 438: please compile the yeast strains into a table for ease of genotype comparison.  Include the specific alleles for all mutations (auxotrophic markers, repair mutants, etc.).

A: We included Table 4, containing information about yeast strains used in the study.

R: 4. Line 436: include details of SC/MD media.

A: We added necessary information:

Cells were grown on either rich YPD media (1% yeast extract, 2% peptone, 2% dextrose) or minimal synthetic media (minimal dextrose, MD) containing Yeast nitrogen base (0.67%) with ammonium sulfate, glucose (2%), agar for solid media (2%) and specific amino acids and nitrogenous bases required for selection.

Minor comments:

R: 5. Line 71: I would suggest adding verbiage to direct the reader to the appropriate section of the figure.  For example, when discussing the damage that accumulates on the DNA refer to the top part of the figure, or label it part A.  Then when discussing the repair/mutagenic repair I would suggest verbiage indicating the bottom left, or label part B.  And when discussing the temporary phenotypic change prior to repair/mutation add verbiage to indicate the bottom right, or label part c.  These are not significant changes, but I think that the more precision that is presented will facilitate readers to the appropriate sections as needed.  Furthermore, the legend should be a bit more descriptive than it currently is written to allow the reader to interpret the sections – and all the various cartoons images used to depict various types of damage that can accumulate.

A: Figure 1 represents a simple scheme of what can happen with DNA primarily lesions. We feel that subdividing will complicate it. We modified the Fig1 legend to clarify it.

R: Line 78: when discussing 8-oxoG and uracil resulting in protein variant production in humans and E. coli, please add a specific example from the works that you cited – from either organism.  (Similar to how you cite the effects on nitrate reductase U. maydis in the previous portion of the paragraph).

A: We add some details to this section.

R: 7. Lines 191-192: The border scheme for the rows that comprise table #1 are a bit distracting as presented.  Please reformat in a more aesthetic, clearer manner.

A: All tables are created according journal requirements and templates, for example. If the editors don't mind, we have added vertical borders inside the table.

R: 8. Line 319: Figure 5E – please reformat the numerical labels on the chart so they do not overlap with different bars (e.g. the WT/untreated vs mutant/mutagen bars).

A: We mowed numerical labels a bit.

R: 9. Line 346-347: Figure 6 legend is too brief and lacks appropriate verbiage and description for the figure.  It requires revision for clarity.

A: We made changes to Fig. 6 legend:

Figure 6. Scheme of yeast cell cycle representing physiological markers for each phase and critical points when yeast cells are capable of mating or the most sensitive to certain DNA damaging agents. Since hybridization in yeast is only possible when both cells are synchronized at the G1 stage, the use of mutagenic factors acting at specific stages of the cell cycle allows to assess the ability of a primary lesion to persist over several stages of the cell cycle. Thus, CPT predominantly causes DNA breaks at the S stage, and then they may be repaired by the homologous recombination system in error-free or error-prone manner at the G2 stage. The repair result can be accounted at the G1 stage by changes in the frequency of illegitimate hybridization and the phenotype of illegitimate hybrids.

R: 10. Line 408: This reads like a conclusion section.  I would recommend making these final two paragraphs of the results and discussion into a conclusion section and re-numbering the following sections accordingly.  **** after completing the materials and methods I found that there is a Conclusion section!  Move this before the materials and methods (it will most likely be missed by many readers otherwise).

A: We used the IJMS template for the manuscript preparation. The current position of Conclusion section corresponds to the journal’s requirements.

R: Grammar:

11. Line 26: missing punctuation – “Spontaneous or induced DNA lesions can due to their inaccurate repair result in stable…” should read: Spontaneous or induced DNA lesions can, due to their inaccurate repair, result in stable…”

 Corrected.

 12. Line 184: Spacing issue after ref 29 and the start of the next sentence.

Corrected.

 13. Line 285-286: “pairing with “C” and od non-canonical pairing with with…” – O believe that the word ‘od’ is a typo. Furthermore the duplicate use of the word ‘with’ appears to be another typo.

Corrected.

14. Line 387: “the cdc28-4 strain has a partial the cell cycle arrest” – should read has a partial cell cycle arrest. Delete the word the.

Corrected.
